# Proof-of-Concept of Detection of Counterfeit Medicine through Polymeric Materials Analysis of Plastics Packaging

**DOI:** 10.3390/polym13132185

**Published:** 2021-06-30

**Authors:** Mohammad Rizalmazli Salim, Riyanto Teguh Widodo, Mohamed Ibrahim Noordin

**Affiliations:** 1Department of Pharmaceutical Technology, Faculty of Pharmacy, Universiti Malaya, Jln Profesor Diraja Ungku Aziz, Kuala Lumpur 50603, Malaysia; ibrahimn@um.edu.my; 2Pharmacy Enforcement Division, Ministry of Health, Malaysia, Jln Profesor Diraja Ungku Aziz, Petaling Jaya 46200, Malaysia

**Keywords:** counterfeit medicine, DSC, FTIR, polymer, blister packaging

## Abstract

The detection of counterfeit pharmaceuticals is always a major challenge, but the early detection of counterfeit medicine in a country will reduce the fatal risk among consumers. Technically, fast laboratory testing is vital to develop an effective surveillance and monitoring system of counterfeit medicines. This study proposed the combination of Attenuated Total Reflectance Fourier Transform Infrared (ATR-FTIR) and Differential Scanning Calorimetry (DSC) for the quick detection of counterfeit medicines, through the polymer analysis of blister packaging materials. A sample set containing three sets of original and counterfeit medicine was analyzed using ATR-FTIR and DSC, while the spectra from ATR-FTIR were employed as a fingerprint for the polymer characterization. Intending to analyze the polymeric material of each sample, DSC was set at a heating rate of 10 °C min^−l^ and within a temperature range of 0–400 °C, with nitrogen as a purge gas at a flow rate of 20 mL min^−1^. The ATR-FTIR spectra revealed the chemical characteristics of the plastic packaging of fake and original medicines. Further analysis of the counterfeit medicine’s packaging with DSC exhibited a distinct difference from the original due to the composition of polymers in the packaging material used. Overall, this study confirmed that the rapid analysis of polymeric materials through ATR-FTIR and comparing DSC thermograms of the plastic in their packaging effectively distinguished counterfeit drug products.

## 1. Introduction

Pharmaceutical packaging is a USD 70 billion global industry with an estimated 6% year-over-year CAGR [1]. Medicine, unlike other packaging categories, is still in demand, and the market is even more essential now, as the COVID-19 pandemic has reached a critical stage.

Pharmaceutical packaging protects medicines from contamination, damage, deterioration and counterfeiting, alongside extending product shelf life. International regulatory bodies have implored drug and packaging manufacturers to play a critical role in the drug supply chain.

Plastics are the most popular material used in the packaging industry [2]. Plastic belongs to one class of polymer known as thermoplastics, which has a range of different properties. The structures of thermoplastics enable polymer chains to move freely, to change forms as a function of temperature [3]. For example, polyethylene, polystyrene, polyvinyl chloride and polypropylene are the common polymers manipulated in the preparation of plastic containers of various densities to fit specific formulation needs.

Growth in the pharmaceutical packaging market is attributed to increased R&D, generics, innovative packaging materials and increased outsourcing to contract packagers. Plastic bottles, parenteral containers, blister packaging, specialty bags, closures, labels and other products are in massive demand in the pharmaceutical industry. However, an increase in packaging costs resulting from strict regulations and anticounterfeiting measures restricts the full growth potential of the market.

Counterfeit medicine is an alarming global problem that affects developing and developed countries with strong regulatory and market regulation structures. This is supported by conservative figures from the last decade, in which 10% of all pharmaceutical products consumed worldwide are counterfeit. This figure could have exceeded 50% in some parts of Africa and Asia [4,5].

All categories of medicines can be and have been counterfeited, including expensive lifestyle medicines, such as drugs for treating erectile dysfunction, fat-lowering, or sleeping pills; antibiotics; anticancer drugs; and medicine for lowering hypertension or cholesterol. Similarly, popular and low-cost medicines are also susceptible to counterfeiting misconduct, namely, painkillers and antihistamines [6,7,8].

A counterfeit medication, according to the World Health Organization (WHO), is “a medicine that is intentionally and fraudulently mislabeled with regard to its identity and source, and includes, but is not limited to, medicines that contain no active ingredient, the wrong amount of active ingredient, the wrong active ingredient, high levels of toxins and false packaging”.

Counterfeiters manufacture counterfeit medicines to trick and confuse medical professionals, patients and consumers by imitating authentic medical products. Due to similar packaging, users are often unable to detect the difference between genuine and fake medicine [9].

Authorities are urged to increase their efforts in establishing rapid identification methods and regulating the market as stern precautions to control/combat widespread counterfeit drugs. The aforementioned efforts are urgently needed in order to protect consumers and pharmaceutical revenues, and avoid the emergence of drug resistance as a result of incomplete dosing schemes [10].

Fortunately, a variety of modern analytical tools have been proven to be successful, such as chromatography and spectroscopy carried out in either the laboratory or in the subject, and have managed to eradicate or curb the issue of distinguishing between counterfeit and genuine products [11,12].

However, the majority of techniques are time consuming and require extensive pattern planning or sample destruction. On this account, improving an analytical approach to simultaneously distinguish legitimate and counterfeit medicine samples from blister packaging tests is critical, as it enables the tablet to remain intact in its physical form [13].

Additionally, a shaping film and a lidding cloth are used to shape blister packaging. The forming film is a packaging material made of polymeric material and a coating agent that encases the product in deep drawn pockets. As for the lidding material, it is composed of rugged temper aluminum, paint primer and a sealing agent. It serves as the foundation or most needed structural element for the final blister packaging. In this study, the polymeric material of the blister film was used as the most crucial packaging material, with the purpose of detecting and investigating counterfeit pharmaceutical products.

Notably, Attenuated Total Reflectance-Fourier Transform Infrared (ATR-FTIR) and Differential Scanning Calorimetry (DSC) are two analytical methods suitable for the forensic study of the polymer film in the blister packaging of medicine. FTIR and DSC can characterize the differences in blister films obtained from both genuine and fake pharmaceutical products. ATR is a common FTIR sampling technique as it does not require samples to be placed in KBr pellets or pretreated. Alternatively, the sample was measured by constructing direct contact with the crystal surface to obtain spectroscopic data. ATR-FTIR was used to determine the polymorphic content of bulk pharmaceutical materials and powder mixtures, and adulteration [14,15]. Several studies have adopted the techniques mentioned above to analyze spectroscopy data in forensic cases, including counterfeit medicines [16,17,18].

Differential Scanning Calorimetry (DSC) is also deemed to have more benefits in assessing content purity than other available techniques. DSC calculates the temperatures and heat flow associated with a material’s thermal transition as a function of time and temperature [19]. It provides qualitative and quantitative data on the physical and chemical changes in heat ability produced by either endothermic or exothermic processes [20]. A comparison of thermograms of the polymeric material in the blister packs between genuine and counterfeit medicines can be a reference to detect counterfeit drugs.

DSC could be treated as a fast counterfeit drug detector to analyze the polymer in drug packaging. DSC has been proven to effectively detect fake medication and the purity of drug portion [21,22,23]. The food packaging industry also uses DSC analysis to assess the plastics’ consistency in the packaging materials [24,25,26]. Combining the benefits of DSC and ATR-FTIR indicates a promising possibility to quickly identify counterfeit drugs in the market, by investigating their polymer material in drug packaging using spectroscopy and thermal analysis. Some analytical methods have been employed for counterfeit medicine analysis; however, there are no related studies that focus on the analysis of polymeric material of the packaging using FTIR and DSC for counterfeit drug identification. The combination technique will expand the analysis field and minimize errors to detect and identify counterfeit medicine in the market.

## 2. Materials and Methods

### 2.1. Materials

Three brands of commonly counterfeited pharmaceutical products in Malaysia were analyzed in this research. Authentic pharmaceutical product samples were collected in five different batches from the licensed pharmacy, and five replicates of counterfeited samples were obtained from the exhibit seized by the enforcement agency. A coding system was applied to identify and recognize samples for analysis. All samples were coded as listed in Table 1.

### 2.2. Method

#### 2.2.1. ATR-FTIR Analysis

This study opted for ATR-FTIR analysis as a tool to identify the type of polymer used for the packaging materials.

ATR-FTIR analysis was conducted using the Bruker ALPHA FTIR Routine Spectrometer. Polystyrene calibration film (0.038 mm) was used to calibrate the spectrometer to ensure that an accurate wavelength was obtained throughout the experiment. We then employed the Attenuated Total Reflectance (ATR) technique to analyze the samples.

The polymer/plastic packaging material from authentic and counterfeit samples was cut into small pieces using stainless steel scissors and placed in the center of the diamond crystal surface of the FTIR. Uniform and constant pressure was applied directly onto the sample on the surface by rotating the pressure device until it stopped at maximum to ensure the attainment of high-quality spectra. The spectra were obtained in the spectral region of 4000–500 cm^−1^ with a resolution of 4 cm^−1^ and were executed in 8 scans. Afterwards, the obtained spectra were analyzed using OPUS software. Identification was achieved by comparing the obtained spectra with Bio-Rad Spectral Database (Bio-Rad, Hercules, CA, USA). The spectra gained for the authentic and suspected counterfeit packaging materials were analyzed and compared to determine counterfeiting elements. The fake and original pharmaceutical products were further analyzed using DSC.

#### 2.2.2. DSC Analysis

Thermal analysis was carried out using the Perkin Elmer DSC 6000, which was connected to a chiller (Intracooler SP, Perkin Elmer, Waltham, MA, USA) and a thermal analysis gas station (Perkin Elmer, Waltham, MA, USA) to control the flow of the purge gas, nitrogen, at a flow rate of 20 mL min^−1^. Indium and zinc (Perkin Elmer, Waltham, MA, USA) were used to calibrate the DSC.

The polymer/plastic packaging material was cut into small pieces and placed in the center of the DSC pan. An amount of about 1 to 1.5 mg was weighed using a microbalance (Mettler Toledo UMT2), placed in an aluminum pan (sealed pan, Kit No. 0219-0062, Perkin Elmer, Waltham, MA, USA), and later, sealed. The DSC was programmed to scan the sample by heating from 0 to 400 °C at 10 °C min^−1^.

The Pyris Manager software was used to determine the onset of melting temperature, peak temperature, end of melting temperature and energy taken for the melting process (∆H).

## 3. Results

The study’s primary focus was on the 1500–500 cm^−1^ fingerprinting regions to characterize and verify the polymer used in the samples obtained. This process enabled the detection of differences in the sample spectra, allowing counterfeit medicine identification. The ATR-FTIR spectra of the counterfeit (A1, B1 and C1) plastic blister samples and the authentic plastic blister samples displayed very similar characteristics (Figure 1). A further comparison between the authentic and counterfeit samples indicated that both characteristics’ peaks stemmed from a particular polymer compound and the regions of spectral overlap (Figure 2, Figure 3 and Figure 4). Table 2 presents the polymer spectra of the counterfeit and authentic samples.

Figure 2 shows typical ATR-FTIR spectra of blister packaging of samples of authentic A and counterfeit A1. The main bands identified in the ATR-FTIR spectra are as follows: 609 cm^−1^ and 679–681 cm^−1^ due to C–Cl bond stretching [27]; 964 cm^−1^: rocking of CH_2_ group; 1251–1252 cm^−1^: C–H bending; 1328–1330 cm^−1^ and 1426 cm^−1^: CH_2_ bending. These results suggest the presence of PVC in the chemical composition of the forming polymer of the blister packaging. Generally, a similar chemical profile of ATR-FTIR spectra was observed between Samples A and A1.

Figure 3 shows the ATR-FTIR spectra of blister packaging for authentic B and counterfeit B1 samples. The ATR-FTIR spectra identified are as follows: 523–524 cm^−1^, 610–612 cm^−1^ and 697 cm^−1^: C–Cl stretching; 964 cm^−1^: rocking of CH_2_ group; 1096 cm^−1^: C–C stretching bond of the PVC backbone chain; 1252 cm^−1^: C–H bending corresponding to Cl group; 1330 and 1426 cm^−1^: CH_2_ bending [28]. Similar to Figure 2, these results also suggest the presence of PVC in both Samples B and B1.

Figure 4 also shows similar ATR-FTIR spectra between Samples C and C1. The spectra for the authentic and counterfeit samples are as follows: 518–523 cm^−1^, 610–612 cm^−1^ and 694–698 cm^−1^: correspondence to C–Cl bond stretching; 962–964 cm^−1^: rocking of CH_2_ group; 1096 cm^−1^: back bone chain of C–C stretching [29]; 1252 cm^−1^: CH bending; 1328–1330 cm^−1^ and 1426 cm^−1^: CH_2_ bending. PVC was also present in samples C and C1 as similar chemical profiles were identified between these samples. Further comparison with the spectral library revealed that the polymer used in the production of both plastic blisters was polyvinyl chloride (PVC). The spectra showed that the counterfeit samples were all forged from the same polymer material and claimed to be the authentic blister pack samples.

Five different batches of authentic samples (A, B and C) and five replicates of the counterfeit samples (A1, B1 and C1) were analyzed to study the polymer characteristics of product packaging. A distinct difference was observable from the DSC thermogram of the genuine and counterfeit products. Thermograms for both Samples A and A1 consisted of two peaks with a similar pattern (Figure 5). The authentic Sample A exhibited two peaks where the polymer was melting at 255.92 ± 1.06 °C and a distinct exothermic peak which was at 293.41 ± 2.12 °C. The DSC thermogram for the counterfeit A1 sample blister pack showed a higher melting point at 267.48 ± 1.23 °C and a higher exothermic peak at 308.03 ± 2.11 °C. These results suggest that the polymer used in counterfeit A1′s plastic blisters was made of a different quality PVC than the original blister (Sample A). This was explained by a difference of only a few degrees in T_m_°, yet this could be significant in establishing basic crystallization mechanisms. Samples B and B1 also exhibited two peaks in their DSC thermogram (Figure 6). The polymer from Sample B, an authentic product, had a lower melting and exothermic peak than B1. It melted at 239.92 ± 1.06°C and also showed a distinct exothermic peak at 287.41 ± 12.59 °C. The DSC thermogram for Sample B1 plastic blisters also consisted of two peaks, but the melting point was at 265.48 ± 2.34 °C, and the exothermic peak was at 299.13 ± 3.88 °C. These results specified a different polymer melting point between Samples B and B1. Samples C and C1, which are counterfeit and original sildenafil plastics, respectively, consisted of two peaks with a similar pattern. Nevertheless, the original sildenafil plastic exhibited two peaks, which were the polymer melting at 271.45 ± 2.12 °C and a distinct exothermic peak at 294.05 ± 5.78 °C (Figure 7). The DSC thermogram for the counterfeit sample plastic blisters (Sample C) also consisted of two peaks but at a higher temperature, 281.23 ± 1.68 °C for the melting point and an exothermic peak at 299.8 ± 1.67 °C. These results further suggest that the polymer used in the Sample C blister pack was different from the polymer used in counterfeit Sample C1. The onset, peak, end of melting and enthalpy change in fusion (∆H) for the DSC analysis are presented in Table 3.

All polymers from the pharmaceutical samples were identified by FTIR analysis, and the spectral library revealed that the polymer used in the production of plastic blisters in all samples was polyvinyl chloride (PVC). DSC has the upper hand in understanding the crystallinity and melting point of a polymer, which is particularly valuable, since PVC has a broad melting point [30]. Even though the same polymer was submitted to the DSC analysis, it was able to distinguish between both polymers using their melting point, heat of fusion and crystallization point. PVC is also recognized to have the ability to essentially adjust its elasticity and hardness through the addition of a plasticizer [31], which makes it possible to alter the melting point of the blister. The counterfeiter may use PVC with a different compatibilizer in their product [32]. Hence, it can be concluded that another quality of PVC was used to produce the counterfeit pharmaceutical plastic blisters, obtainable from a different supplier or by using a compatibilizer that is different from the legitimate manufacturer of the authentic medicine [33]. As demonstrated in this study, DSC allows differentiating between different qualities of PVC solely based on their thermal transitions, despite having a similar FTIR spectrum. Changes such as curing and crystallization may reveal the difference in the manufacturing processes by different manufacturers. Different polymers, such as polyethylene (LDPE and HDPE), were also easily distinguishable based on their melting point using DSC [34], even though the FTIR spectrum exhibited a similar blueprint. Moreover, DSC excels in terms of observation of the thermal transitions of a polymer blend. FTIR analysis, on the other hand, requires trained personnel to interpret the result. It focused on identifying characteristic spectral bands with the analysis of the registered spectra to identify the functional groups of active compounds introduced to the polymer material.

## 4. Conclusions

This work substantiated that ATR-FTIR and DSC are powerful analytical tools for polymer identification. The ability of ATR-FTIR to characterize the polymer used to discern between authentic and counterfeit samples shows some limitations if a polymer with the same chemical compound is used in the packaging materials. DSC demonstrated its ability to quickly detect fake products by differentiating the polymers’ melting points in both samples. DSC allowed us to distinguish between different PVC qualities based on their thermal transitions, despite having a similar FTIR spectrum. Conducting a forensic analysis on the polymer material using FTIR and DSC ensures vital information reaches the relevant authorities, who will pursue this in a continuous attempt to scrutinize counterfeit medicinal product sources. Identifying various substances can provide leads for investigators who can subsequently confirm or refute their suspicions. Finally, the data of this study established that counterfeit medicine could be detected by analyzing the polymer materials in the packaging. This will provide aid to law enforcement as well as other members of the criminal justice and legal systems, with the intention to successfully investigate and adjudicate these crimes.

## Figures and Tables

**Figure 1 polymers-13-02185-f001:**
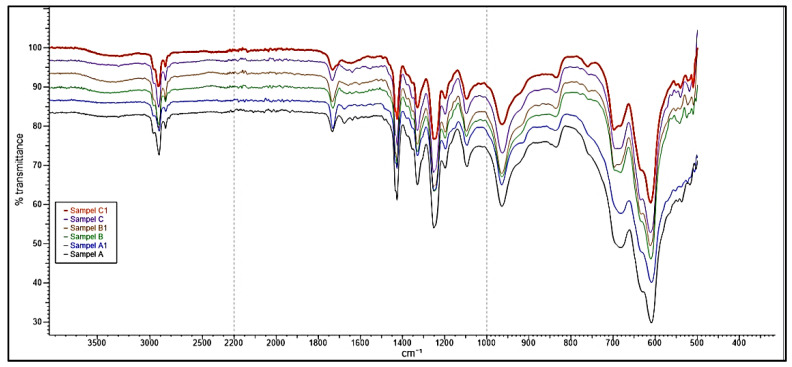
ATR-FTIR spectra of blister packaging for samples A, A1, B, B1, C and C1.

**Figure 2 polymers-13-02185-f002:**
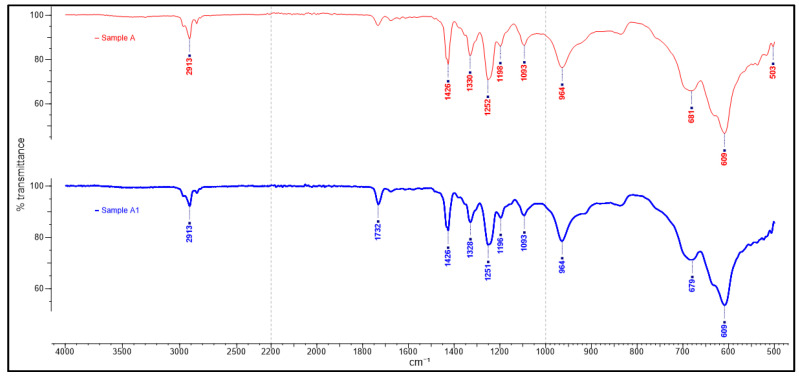
ATR-FTIR spectra of blister packaging for samples A and A1.

**Figure 3 polymers-13-02185-f003:**
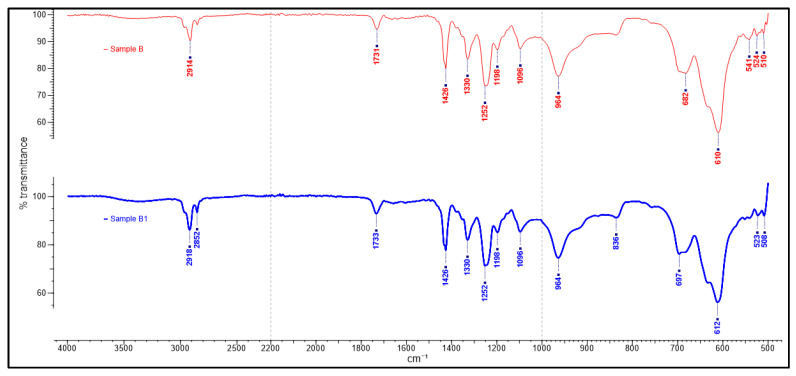
ATR-FTIR spectra of blister packaging for samples B and B1.

**Figure 4 polymers-13-02185-f004:**
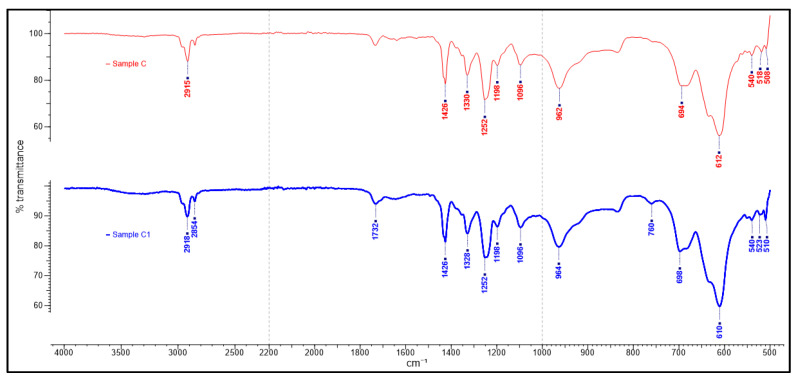
ATR-FTIR spectra of blister packaging for samples C and C1.

**Figure 5 polymers-13-02185-f005:**
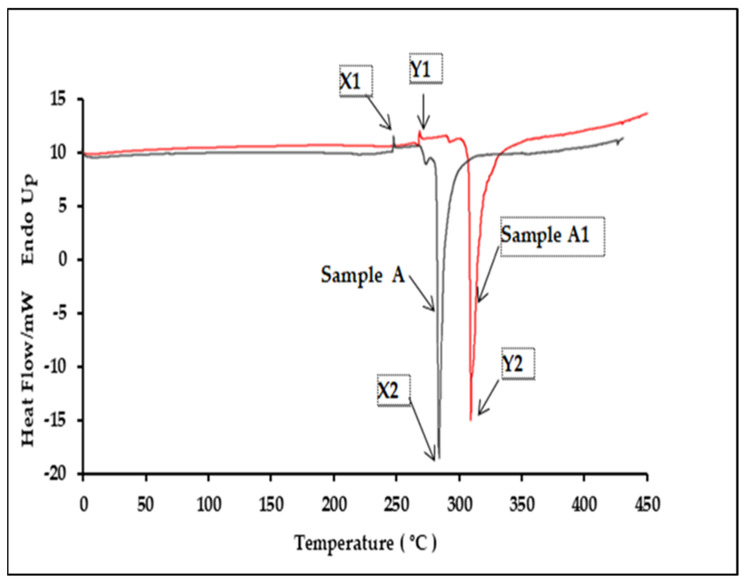
Comparison of the DSC thermograms between the authentic and counterfeit samples, A and A1.

**Figure 6 polymers-13-02185-f006:**
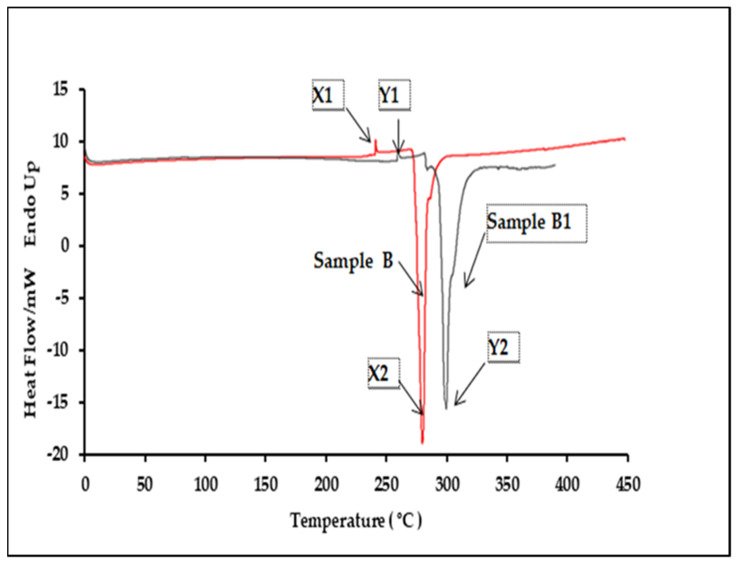
Comparison of the DSC thermograms between the authentic and counterfeit samples, B and B1.

**Figure 7 polymers-13-02185-f007:**
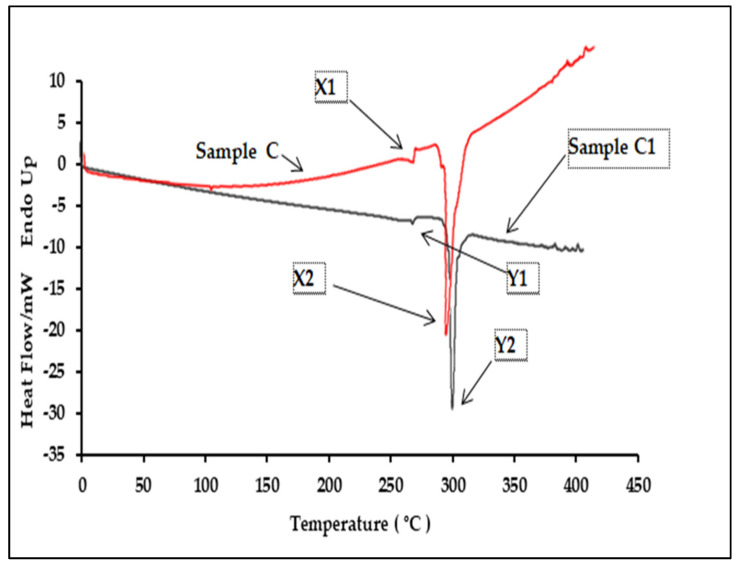
Comparison of the DSC thermograms between the authentic and counterfeit samples, C and C1.

**Table 1 polymers-13-02185-t001:** Description of samples for ATR-FTIR and DSC analysis.

Code	Sample Type	Description
A	Original	Blister of 8 × 500 mg paracetamol caplets
A1	Counterfeit	Blister of 8 × 500 mg paracetamol caplets
B	Original	Blister of 6 × 500 mg paracetamol caplets
B1	Counterfeit	Blister of 6 × 500 mg paracetamol caplets
C	Original	Blister of 4 × 100 mg sildenafil tablet
C1	Counterfeit	Blister of 4 × 100 mg sildenafil tablet

**Table 2 polymers-13-02185-t002:** Infrared characteristics and wave number of authentic and counterfeit pharmaceutical products.

Product 1				Product 2				Product 3			
Sampel A		Sampel A1		Sampel B		Sampel B1		Sampel C		Sampel C1	
Wave Number (cm^−1^)	Functional Group	Wave Number (cm^−1^)	Functional Group	Wave Number (cm^−1^)	Functional Group	Wave Number (cm^−1^)	Functional Group	Wave Number (cm^−1^)	Functional Group	Wave Number (cm^−1^)	Functional Group
503	C–CI	503	C–CI	524	C–CI	523	C–CI	518	C–CI	523	C–CI
609	C–Cl	609	C–Cl	610	C–Cl	612	C–Cl	612	C–Cl	610	C–Cl
681	C–Cl	679	C–Cl	682	C–CI	697	C–CI	694	C–Cl	698	C–Cl
964	CH_2_	964	CH_2_	964	CH_2_	964	CH_2_	962	CH_2_	964	CH_2_
1093	C–C	1093	C–C	1096	C–C	1096	C–C	1096	C–C	1096	C–C
1252	CH	1251	CH	1252	CH	1252	CH	1252	CH	1252	CH
1330	CH_2_	1328	CH_2_	1330	CH_2_	1330	CH_2_	1330	CH_2_	1328	CH_2_
1426	CH_2_	1426	CH_2_	1426	CH_2_	1426	CH_2_	1426	CH_2_	1426	CH_2_
2913	CH_2_	2913	CH_2_	2914	CH_2_	2918	CH_2_	2915	CH_2_	2918	CH_2_

**Table 3 polymers-13-02185-t003:** The onset, peak, end of melting and enthalpy change in fusion for the authentic and the counterfeit samples, A, A1, B, B1, C and C1.

Product		The Onset of Melting (°C)	The Peak of Melting (°C)	End of Melting (°C)	∆H (J/g)
A	X1	254.52 ± 2.73	255.91 ± 1.06	256.682 ± 1.20	7.039 ± 1.74
	X2	289.70 ± 2.29	293.40 ± 2.12	298.35 ± 2.08	1168.51 ± 198.83
A1	Y1	267.09 ± 1.34	267.47 ± 1.23	268.77 ± 1.22	6.52 ± 1.31
	Y2	304.03 ± 4.00	308.02 ± 2.11	313.74 ± 1.56	1254.20 ± 219.37
B	X1	239.45 ± 1.08	239.92 ± 1.06	241.48 ± 1.10	8.74 ± 1.70
	X2	283.17 ± 12.6	287.18 ± 12.01	291.91 ± 12.61	1372.09 ± 335.71
B1	Y1	264.00 ± 2.13	265.23 ± 2.23	274.57 ± 15.15	4.19 ± 1.14
	Y2	295.06 ± 4.00	299.13 ± 3.88	302.74 ± 4.62	1188.72 ± 240.41
C	X1	268.14 ± 1.08	271.45 ± 2.12	273.27 ± 2.24	7.23 ± 1.70
	X2	290.62 ± 5.88	294.05 ± 5.78	298.74 ± 5.75	674.60 ± 129.12
C1	Y1	281.23 ± 3.01	283.59 ± 1.68	284.89 ± 2.75	8.23 ± 1.44
	Y2	297.55 ± 1.41	299.79 ± 1.67	302.53 ± 2.12	488.03 ± 143.26

## Data Availability

The data presented in this study are available on request from the corresponding authors.

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
