# Peer review of "Proof-of-Concept of Detection of Counterfeit Medicine through Polymeric Materials Analysis of Plastics Packaging"

_polymers, 2021, doi:10.3390/polym13132185_

Round 1
Reviewer 1 Report
The reader is well introduced to the subject and only in a few sentences some references should be increased (lines 98, 108, 109).
In line 79, I would suggest change "while enabling" with "as it enables".
In line 104, I would suggest "A comparison of thermograms of the polymeric...".
In paragraph 2.2.1 ATR-FTIR Analysis, please add the spectral resolution.
Figures 2-4: as ATR on blister revealed itself to be not useful to distinguish between authentic and counterfeit samples and as all the spectra are very similar, I would suggest to remove these figures and add the peak marks to figure 1.
Table 2: are the authors confident two decimal places to be appropriate with the given number of scans, spectral resolution and, in a more general view, with the technique employed? I would suggest to round off the numbers to the unit.
Figures 5-7: what are the curves inside the figures? Specify if are the average curves or other. Moreover, please add Heat flow scale and connectors between X/Y marks and peaks. Finally, in figure 7 specify which curve is C1.
Lines 185-207: if data are in the table 3 these lines are not useful. The authors should comment extensively the possible interpretation of such differences and their causes (in the same way in lines 221-231).
Table 3: some uncertainties do not match the decimal places of the related measurement: please check the values. For true, uncertainties with more than one significant value are considered extremely precise and here Table 3 even shows three significant values: the authors must report uncertainties with one or at least two values and adjust the related decimal places in measurements values.
Line 219: please remove "as PVC", as it is explained at the end of the sentence.
In Conclusions, lines 233-235: I suggest to remove these lines as the FTIR analysis is obviously able to recognize the major component of the blister plastic. I would rather comment how the different sensitivity and LOD of the two techniques are appropriate or not to the detection of slight differences in the composition of the plastic (and put this part in the Results).
Finally, as an overview, I strongly suggest to change the title of this work in "Proof-of-concept of detection of counterfeit medicine through polymeric materials analysis of plastics packaging". It would be much more appropriate as the experimental part is poor with respect to the purpose and no statistical approach was adopted aiming to demonstrate the differences are not due to chance discrepancies between authentic and counterfeit. The work is in a promising direction but would greatly benefit from taking into account collecting and analysing a much larger set of different couples authentic/counterfeit samples.
Reviewer 2 Report
The present manuscript entitled “Detection of Counterfeit Medicine through Polymeric Materials Analysis of Plastics Packaging” authored by Rizalmazli Bin Salim et al. describes a polymeric materials rapid analysis through ATR-FTIR and DSC thermograms of the plastic in their packaging efficiently distinguished counterfeit drug products. Furthermore, the current study clearly indicates the DSC and ATR-FTIR, a promising possibility to quickly identify counterfeit drugs in the market, by investigating their polymer material in drug packaging using spectroscopy and thermal analysis. It is a well-organized article and the study is sufficiently performed. The study is very accurate and adequate, lacks of major errors, and thus, I recommend it for publication. However, certain Minor issues are detailed below which need to be addressed before its final acceptance in the Polymers.
I advise the authors to take the following points into account while revising their manuscript.
Comment 1: There are so many typographical errors in the manuscript text , so authors need to correct it in the revised manuscript. For e.g. Line 18, 10 °C min−1 should be 10 °C min−1, Line 19, 20 ml min−1 should be 20 ml min−1, Line 135, 4000-500 cm-1 should be 4000-500 cm-1 and Line 197, 265.48±2.34°Cand should be 265.48±2.34°C and etc.,
Comment 2: Line 13, “Fourier Transform Infrared (ATR-FTIR)” should be Attenuated Total Reflectance-Fourier Transform Infrared (ATR-FTIR).
Comment 3: Introduction is well written, appropriate information is provided. However, include some more recent years literature in the introduction section to strengthen their work and also before the last paragraph, it is important to add a gap sentence where the authors broadly explain what the novelty of their work has not been studied/reported before.
Comment 4: Figures 1 quality is not good, so provide the high-resolution Figure 1, and Figures 5 and 6 axis titles font is not similar and are currently quite disorganized. So, arrange the axis titles in a proper way and maintain the consistency.
Comment 5: In table 2, Wave Number (Cm-1) should be Wavenumber (cm-1)
Comment 6: The ATR-FTIR results explanation should be discussed wider and compared with the other studies.
Author Response
Dear Reviewer 2,
Please see the attachment. Thank you.
